# Gill Junction Injury and Microbial Disorders Induced by Microcystin-Leucine Arginine in *Lithobates catesbeianus* Tadpoles

**DOI:** 10.3390/toxins14070479

**Published:** 2022-07-13

**Authors:** Huiling Jiang, Jun He, Hui Wang, Lingling Zheng, Xiaoran Wang, Huijuan Zhang, Hailong Wu, Yilin Shu

**Affiliations:** 1Collaborative Innovation Center of Recovery and Reconstruction of Degraded Ecosystem in Wanjiang Basin Co-Founded by Anhui Province and Ministry of Education, School of Ecology and Environment, Anhui Normal University, Wuhu 241002, China; jiangjiangbei@163.com (H.J.); hejun@ahnu.edu.cn (J.H.); shangyuan19971020@163.com (H.W.); zllstxzxc@163.com (L.Z.); misakipaki@163.com (X.W.); huijuan2545@ahnu.edu.cn (H.Z.); 2State Key Laboratory of Freshwater Ecology and Biotechnology, Institute of Hydrobiology, Chinese Academy of Sciences, Wuhan 430072, China

**Keywords:** gill barrier, gill microbiota, MC-LR, hypoxic stress, amphibian

## Abstract

Microcystin-LR (MC-LR) is widely present in waters around the world, but its potential toxic effects and mechanisms on amphibian gills remain unknown. In the present study, tadpoles (*Lithobates catesbeianus*) were exposed to environmentally realistic concentrations of 0.5, 2 μg/L MC-LR, and 0 μg/L MC-LR (Control) for 30 days with the objective to unveil the impairment of gill health. The lysozyme was downregulated, while pattern recognition receptors and complement and adaptive immune processes were upregulated and the ability of gill supernatant to inhibit pathogenic bacteria decreased in the 0.5 and 2 μg/L MC-LR groups. The transcriptions of epithelial barrier components (e.g., *CLDN1*) were significantly decreased in MC-LR-exposed gills, while the gill content of lipopolysaccharide (LPS) endotoxins and the transcriptions of downstream responsive genes (e.g., *TLR4* and *NF-**κB*) were concurrently increased. In addition, the number of eosinophils and the expression of pro-inflammatory cytokines (e.g., IL-1β and *TNF-α*) were increased. These results imply that exposure of tadpoles to low environmentally concentrations of MC-LR leads to inflammation, increased permeability, and a reduced ability to inhibit pathogenic bacteria. The epithelial cells of inner gill filaments increased and transcriptions of hypoxic stress genes (e.g., *HIF-1α*, *FLT1*, and *SERPINE1*) were upregulated within the exposed group. As a consequence, exposure to MC-LR may lead to hypoxic stress. MC-LR exposure also drove gill microbiota to a dysbiosis. The relative abundance of *Elizabethkingia* was positively correlated with content of LPS and transcriptions of *NF-κB* and *TNF-α*. Overall, this study presents the first evidence about the pronounced impacts of MC-LR exposure on gills of amphibians, highlighting the susceptibility of early developing tadpoles to the environmental risks of MC-LR.

## 1. Introduction

In early developing tadpoles and bony fish, the gills have various functions such as gas exchange, immune defense, acid-base balance, and ion regulation [1,2,3]. Gills are usually sensitive to chemicals in water because they are directly and constantly exposed to chemicals dissolved in the water [4]. Chlorpyrifos has been reported to induce damage to the physical and immune barriers of *Cyprinus carpio Linnaeus* gills, while copper oxide nanoparticles can induce a functional respiratory obstacle in Indian freshwater mussels [5,6]. Usually, some organic pollutants can enter other tissues through the gills and accumulate in different tissues [7,8]. The gill mucosa of amphibian tadpoles provides colonization sites for environmental microorganisms during the early stages of development and has evolved through long-term adaptation to form a synergistic symbiosis with the gill mucosa [9]. Gill symbiotic microorganisms are not only involved in regulating gill immune homeostasis but also prevent pathogenic bacteria from infecting gill tissues through competitive or antagonistic effects [10,11]. The effects of contaminants on gill symbiotic microbes have only sporadically been reported. For example, suspended sediment exposure resulted in an increased abundance of potentially pathogenic bacteria in the gill microorganisms of *Amphiprion ocellaris*. Cnidarian biofouling leads to changes in the composition and structure of gill microbial communities in *Salmo salar* [11,12]. However, the effects of environmental pollutants on the physical, immune- and symbiotic microbiota barriers in tadpole gills and the resistance of immunity material to pathogenic bacterial invasion are currently unknown, and revealing these would provide new toxicological evidence for assessing the extent of pollutants on amphibians.

Cyanobacterial blooms are one of the major environmental problems [13,14,15]. *Microcystis* is the most common dominant species in cyanobacterial blooms, and its secondary metabolites, microcystins, are the most widely distributed and harmful cyclic peptide algal toxins in eutrophic waters [16,17]. To date, 279 variants of microcystins have been identified in freshwater systems. Among them, microcystin-leucine arginine (MC-LR) is the most widely distributed in freshwater, accounting for 46% to 99.8% of total microcystins [18,19]. A survey of 134 lakes in New Zealand found that the highest concentration of microcystins was 6.4 μg/L [20]. In 29 pond/ditch drinking water samples collected in Nanning District, Guangxi, China, the average MC-LR concentration was 0.5 μg/L [21]. The microcystin content was 13.38 μg/L, and the highest concentration was 35.8 μg/L in Lake Taihu of China after the collapse of cyanobacterial blooms, which greatly exceeded the WHO standard of 1 μg/L [22]. MC-LR was detected in 9 reservoirs in eastern Cuba, with an average concentration of about 2 μg/L [23]. Both lakeside areas of large bodies of water and small bodies are the habitats of frogs, which are also areas often accompanied by algal toxin distribution [24]. MC-LR is enriched and amplified in multispecies, bringing about endocrine disruption, reproductive disorders [25], cerebral neurological dysfunction [26], and the disruption of lipid metabolism in the intestines [15,27]. Amphibians have highly permeable skin, an amphibious life history, and limited locomotor and migratory capabilities. These characteristics make amphibians extremely sensitive to environmental changes, which thus have become an important model taxon for identifying and assessing the toxic effects of environmental pollutants [28,29]. However, in spite of the increasing pollution and potent toxicities of MC-LR in aquatic ecosystems, there is a scarcity of toxicological knowledge regarding the adverse effects of MC-LR on amphibians [14,30,31,32]. The gill mucosal barrier of tadpoles in early development plays an important role in their growth and development [11,33]. Few toxicological studies have focused on the effects of MC-LR contamination on the gills of aquatic organisms [5,34], especially in amphibians. Recent studies have reported that 0.5 μg/L MC-LR have led to skin and intestinal barrier obstacles and commensal microbial disturbances in *Lithobates catesbeianus* tadpoles [15,30]. However, it is still unclear whether and how low environmentally realistic concentrations of MC-LR exposure will interrupt the gill health of amphibians.

With objectives to elucidate the disrupting effects of MC-LR on the gill epithelial barrier, commensal microbiome, and breathing capacity, the present study performed the exposure tests using tadpoles (*Lithobates catesbeianus*) at the G26 stage (staging standards refer to Gosner stages [35]) at 0.5, 2 μg/L MC-LR (environmentally realistic concentration), and 0 μg/L MC-LR (Control) for 30 days. After exposure, the impairment of gill barrier integrity and function by environmental realistic concentrations of MC-LR was determined by analyzing histology, oxidative stress- and immune-related biochemical indices, transcriptome, and commensal microbiota. Tadpole gills exposed to MC-LR were examined for gene transcriptions and enzyme activities involved in cellular apoptosis and LPS endotoxins perception to provide mechanistic clues. Eventually, the capability of gill extracts to suppress pathogen proliferation was estimated.

## 2. Results

### 2.1. Histopathological Changes in the Gills

Based on the HE method, the effects of different concentrations of MC-LR on gills for histopathological observations are shown in Figure 1 (Figure 1A–C). The number of internal gill filament epithelial cells and eosinophils showed a concentration-dependent increase in MC-LR exposure groups (Appendix A).

### 2.2. Transcriptome Analysis

A total of 406,733,698 clean reads and 22,224 genes from 9 RNA libraries were obtained (Appendix A). Based on the quantified data for all transcriptions, principal component analysis (PCA) indicated that three groups exposed to environmentally realistic concentrations of MC-LR had changes on the transcription levels in gill tissues (Appendix A). For the 0.5MCG and 2MCG, 706 and 666 DEGs were identified, respectively (Appendix A). In the 0.5MCG, 388 GO terms were markedly enriched (*p* < 0.05), including oxygen transport, innate immune response, complement activation, removal of superoxide radicals, and apoptotic processes (Appendix AA). Twelve KEGG pathways were significantly enriched (*p* < 0.05), including phagosomes, herpes simplex infection, and intestinal immune networks for IgA production (Appendix A Appendix A). In the 2MCG, 281 GO terms were markedly enriched (*p* < 0.05), including innate immune response, positive regulation of B cell activation, inflammatory response, oxygen transport, and toll-like receptor signaling pathways (Appendix A Appendix A). Eighteen KEGG pathways (*p* < 0.05), including tight junctions, necroptosis, and complement and coagulation cascades were identified (Appendix A Appendix A).

#### 2.2.1. DEGs Associated with Gill Tight Junctions

In the 0.5MCG, 6 DEGs associated with tight junctions (*MYH13*, *MYH8*, *MYH4*, *ITGB1*, *TUBA1A,* and *CLDN1*) were significantly downregulated (Figure 2A). Eight DEGs associated with tight junctions (*MYH13*, *PDZD2*, *ARHGEF18*, *MYH4*, *MYH10*, *BVES*, *MYL2*, and CLDN1) were markedly downregulated (Figure 2B) in the 2MCG.

#### 2.2.2. DEGs Associated with Inflammation

In the 0.5MCG, 3 DEGs related to inflammasome complex (*NLRP3*, *NLRP12*, and *NLRP4E*), 3 DEGs associated with cytokines (*CCL20*, *XCL1*, and *ZAP70*), and 6 DEGs (*TLR4*, *IRAK4*, *RIPK1*, *IKKA*, *NF-κB*, and *TNF-α*) associated with TLR4/NF-κB signaling pathway were significantly upregulated (Figure 2C). In the 2MCG, *NLRP3* associated with the inflammasome complex, *IL17C* associated with pro-inflammatory, and 2 DEGs associated with chemokines (*XCL1* and *CCL20*) along with the 6 DEGs of the TLR4/NF-κB signaling pathway (*TLR4*, *IRAK4*, *IRAK1*, *IKKA*, *NF-κB*, and *TNF-α*) were significantly upregulated (Figure 2D).

#### 2.2.3. DEGs Associated with Immunity

Gill mucosa is an important immune barrier. In the 0.5MCG, *LYZ1* associated with lysozyme was significantly downregulated, whereas 3 DEGs (*C5*, *C3*, *CFB*) related to complement, 6 DEGs (*NLRP3*, *NLRP4E*, *NLRP12*, *NLRP8*, *TLR5*) with pattern recognition receptors, and 16 DEGs involved in specific immune processes (e.g., *HLA-B*, *CD8A*, *ZAP70*, *IGKV4-1*, and *IL17C*) were significantly upregulated (Figure 3A). In the 2MCG, *LYZ1* associated with lysozyme was significantly downregulated. 4 DEGs (*C3*, *C9*, *CFB*, and *CRP*) related to complement, 3 DEGs (*TLR5*, *NLRP3*, and *NLRP12*) related to pattern recognition receptor, and 12 DEGs in association with specific immunity (e.g., *MHCII-DRA*, *CD8A*, *IGH* and *IL36RN*) were significantly upregulated (Figure 3B)

#### 2.2.4. DEGs Associated with Hypoxic Stress and Oxygen Transport

The HIF-1α signaling pathway is the main pathway of hypoxia stress. Three DEGs (*FLT1*, *SERPINE1*, and *HIF-1α*) in the HIF-1α signaling pathway were upregulated in the 0.5MCG. Four DEGs (*FLT1*, *SERPINE1*, *HMOX1,* and *HIF-1α*) in the HIF-1α signaling pathway were also upregulated in the 2MCG. This indicated that MC-LR pollution resulted in hypoxia stress in the gills. Therefore, the oxygen transport capacity was further analyzed. In the 0.5MCG, there was a significant downregulation of *HBA5* related to oxygen transport, and 2 DEGs (*HBB2* and *HBA3*) were downregulated in the 2MCG (Figure 4A,B). This suggested that MC-LR led to a reduced ability of respiratory capacity in the gill.

#### 2.2.5. DEGs Related to Oxidative Stress and Apoptosis

In the 0.5MCG, 4 DEGs (*DMGDH*, *DHRS7C*, *MSRB2*, and *WWOX*) related to antioxidants were significantly downregulated, and 3 DEGs (*TXNIP*, *AKR1C4*, and *MPO*) associated with oxidation were significantly upregulated (Figure 4C). In the 2MCG, 2 DEGs (*NQO1* and *DUOX2*) related to antioxidants were significantly downregulated, while 2 DEGs (*ALOXE3* and *MPO*) related to oxidation were significantly upregulated (Figure 4D). Usually, oxidative stress leads to apoptosis. *CASP7*, *CASP3*, *FOS*, and *GZMB* were upregulated in the 0.5MCG and 2MCG (Figure 4C,D).

### 2.3. Biochemical Indicators of Gills and Antibacterial Activity

Inspired by the analysis of the transcriptomic data, some biochemical parameters were further analyzed, including gill immune function, anti-pathogenic activity, oxidative stress, apoptosis, endotoxin, and inflammatory factors. Compared with the control, the concentrations of immunoglobulins (IgA, IgG, and IgM) were significantly upregulated and the catalytic activity of LZM was significantly downregulated in the 0.5MCG and2MCG, which was consistent with the transcriptomic data (Figure 5). In the disc diffusion assay, the area of inhibition of pathogenic bacteria (*Escherichia coli*, *Pseudomonas aeruginosa*, and *Staphylococcus aureus*) decreased with increasing concentration in the MC-LR exposure groups, suggesting that MC-LR exposure caused a reduction in the ability of gills to inhibit the growth of pathogenic bacteria (Figure 5). The MC-LR groups significantly (*p* < 0.05) reduced antioxidant enzyme activities, i.e., SOD and GSH levels, while increasing ROS and caspase 3 activity in gill tissue in a concentration-dependent manner relative to the control group (Figure 6). Furthermore, MC-LR significantly increased the levels of LPS and IL-1β in gills relative to the control group (Figure 6).

### 2.4. Dysbiosis of the Gill Microbiome

Gill microbiota is an integral part of maintaining gill health. 16S rRNA gene sequencing was used to profile the changes in the gill microbiota following exposure to different concentrations of MC-LR to assess the effect of MC-LR on gill health. The gill microbiomes were dominated by five dominant phyla, including Bacteroidetes (11.18–63.63%), Firmicites (1.23–48.56%), Fusobacteriota (2.5–15.1%), Proteobacteria (2.4–14.2%), and Desulfobacterota (0.11–1.67%), whereas *Elizabethkingia*, *Bacteroides*, *Succinispira*, *Cetobacterium*, *unclassified-f-Lachnospiraceae*, *norank-f-Peptostreptococcaceae,* and so on were dominant at the genus level (Figure 7A,B). Analysis of alpha-diversity indices (e.g., Sobs, Chao, Shannon, and Simpson) revealed a shift in species richness and diversity due to MC-LR exposure (Appendix A). In addition, remarkable differences in bacterial community composition were observed in MC-LR exposure groups (ANOSIM results: R = 0.726, *p* = 0.001), and the PCoA analysis confirmed the above results (Figure 7C).

Interestingly, at the genus level, there were remarkable changes in the relative abundances of bacteria. Compared with the 0MCG, genera of *unclassified-f-Lachnospiraceae*, *norank-f-Peptostreptococcaceae*, *Succinispira*, *Lachnoclostridium*, *Acetobacterium*, and *Robinsoniella* were markedly decreased in the 0.5MCG (Appendix A), and 2 μg/L MC-LR exposure induced a remarkable decrease in the relative abundance of the genus *unclassified-f-Lachnospiraceae*, *Succinispira*, *Acetobacterium*, *Macellibacteroides*, and *norank-f-norank-o-Clostridia-UCG-014*. (Appendix A). Furthermore, compared with the 0.5MCG, there was a prominent increase in the relative abundance of the genus *Desulfovibrio*, *unclassified-f-Lachnospiraceae*, *Parabacteroides*, and *Succinispira* in 2MCG (Appendix A).

### 2.5. Correlation between Gill Microbiota and Inflammatory Events Triggered by LPS

To assess the potential relationship between inflammatory events triggered by LPS and gill microbiota genera, Spearman analysis was performed to explore any correlation. As shown in Figure 8, relative abundances of *Elizabethkingia* were significantly and positively correlated with inflammatory events triggered by LPS, including the concentrations of LPS and IL-1β, and the transcript levels of *IRAK4*, *IKKA*, *NF-κB*, and *TNF-α*. However, there were significantly negative correlations of *Elizabethkingia* with the transcript level of *CLDN1*, which represents the gill epithelial barrier component.

## 3. Discussion

In the present study, exposure to MC-LR resulted in increase in the permeability of the gill epithelium barrier, hyperplasia of inner gill filaments, impaired respiratory function, immune dysregulation, and inhibited antimicrobial capacity. MC-LR subchronic exposure also drove the gill microbiome to dysbiosis. In addition, oxidative stress-mediated apoptosis and increased lipopolysaccharide (LPS) endotoxin can lead to the activation of the TLR4/TNF-α signaling pathway, which is an important regulatory mechanism of gill injury after subchronic exposure to MC-LR. The present findings highlighted the susceptibility of early developing tadpoles to MC-LR, which needs an urgent and accurate evaluation.

The gill epithelial barrier separates the organism from its environment and protects it from pathogens and toxins [36]. Tight junctions are a major component of the gill epithelial barrier [37]. The disruption of the tight junctions leads to increased permeability and the penetration of exogenous substances into the body inducing local or systemic inflammatory response [38,39]. After MC-LR exposure, the transcription levels of tight junction genes were downregulated, implying that real environmental concentrations (0.5 μg/L and 2 μg/L) resulted in an impaired gill epithelial barrier. A previous study has shown that higher exposures of 10 μg/L MC-LR impaired the gill tissue barrier in *Cyprinus carpio Linnaeus* [5]. The implication is that MC-LR exposure can leads the gill tissue barrier of amphibian tadpoles more susceptible to damage than some fish. In addition, eosinophils can promote the progression of inflammation by the release the contents of granules [40]. There was a significant concentration-dependent increase in eosinophils in the gill tissue of the MC-LR exposed group. The transcription of genes associated with inflammation was also upregulated in the MC-LR group, manifesting as an enhanced inflammatory response. These results emphasize is similar to previous observations about MC-LR exposure in *L. catesbeianus* gut tissue [14]. MC-LR usually induces oxidative stress in different tissues, such as gonad and liver [31,37]. The oxidative stress was also observed in tadpole gills after exposure to low environmental concentrations of MC-LR. Furthermore, oxidative stress can usually activate the caspase cascade reaction and apoptosis [41,42,43]. The caspase 3 activity and other related apoptotic genes transcription were upregulated after exposure to low environmental concentrations of MC-LR. These results emphasize that MC-LR exposure produces oxidative stress to promote apoptosis, which is similar to previous observations about MC-LR exposure in *Rana nigromaculata* liver and gonad tissue [31,44]. However, no oxidative stress was observed in *Xenopus laevis* tadpoles [45], suggesting that there may still be differences in MC-LR sensitivity in different amphibians.

The gill mucosa is an important immune barrier, including innate and adaptive immunity [46,47]. Lysozyme is the widely distributed enzyme of the innate immune system that exhibits antibacterial activity [48]. After MC-LR exposure, the expression of lysozyme was reduced in gill tissue, while other nonspecific and specific immunities were upregulated. The disc assay can assess immune substances in tissues on the ability to inhibit bacteria after contaminant exposure [30,49]. The disc diffusion test showed that MC-LR exposure induced a reduction in the ability of the gills to inhibit the growth of pathogenic bacteria. Therefore, altered immune material following MC-LR exposure affects gill ability to inhibit pathogens, further implying that gill tissue is also more susceptible to pathogenic bacteria following MC-LR exposure.

Early developmental tadpoles exchange gas primarily through the gills. After exposure, the epithelial cells of inner gill filaments increased with increasing concentrations of MC-LR. HIF-1α signaling pathway genes associated with hypoxic stress were upregulated, implying the presence of hypoxic stress after MC-LR exposure. Generally, hypoxic stress implies a reduction in oxygen transport capacity [50], and a similar phenomenon of reduced oxygen transport capacity was seen in tadpole gills with MC-LR exposure. Thus, realistic environmental concentrations of MC-LR exposure induce the proliferation of gill filament epithelial cells, which may thicken gill filament, separate the blood from the air, produce hypoxic stress, and reduce the oxygen transport capacity. However, the mechanism of gill filament epithelial cell proliferation is yet to be further studied after environmental pollutants exposure [5].

The gill microbiome, together with the gill, serves as a barrier for the host to adapt to external environments and maintain its health [10,11,51]. LPS constitutes a significant portion of the cell wall of gram-negative bacteria [52]. A significant increase in LPS endotoxin content was found in the gills of tadpoles exposed to MC-LR. The correlation analysis indicated a significant positive correlation between LPS content and relative abundances of *Elizabethkingia* genus that is Gram-negative. This implies that *Elizabethkingia* may be an important source of LPS in the gill tissue after MC-LR exposure. LPS infiltration activated *TLR4* and induced the key inflammatory transcript factor *NF-κB* to translocate into the nucleus, leading to the upregulation of a variety of pro-inflammatory cytokines, such as IL-1β and *TNF-α* [53,54]. Activated TLR4/NF-κB signaling pathways and inflammation were also observed in several animal models and cell lines exposed to microcystins [44,55,56,57]. In this study, a significant increase in LPS content in the gill was found after MC-LR exposure, and the transcriptions of *TLR4* and *NF-κB* were also upregulated. In addition, expression of the pro-inflammatory cytokines IL-1β and *TNF-α* was significantly elevated in the gills after MC-LR exposure. Therefore, LPS infiltration activated the TLR4/TNF-α signaling pathway may be an important pathway to cause gill inflammation. MC-LR exposure at environmentally realistic concentrations altered the composition and decline diversity of the gill microbial community. Significantly reducing the diversity of gill microbiota may lead to a reduction in competition or ability to fight pathogens [58]. *Acetobacterium, Succinispira, Lachnoclostridium*, *Eubacterium*, and *Desulfovibrio* bacteria have the function of maintaining the immune barrier [59,60,61,62,63]. The significant reduction of related genera in the gill of MC-LR-exposed tadpoles may mean that MC-LR exposure further interferes with gill mucosal immunity. Norank Clostridia-UCG-014 and *Robinsoniella* bacteria have potential anti-inflammatory effects [64,65]. The significant reduction of norank Clostridia-UCG-014 and *Robinsoniella* bacteria after exposure in MC-LR indicates fluctuations in abundances of norank Clostridia-UCG-014 and *Robinsoniella* may not be conducive to the elimination of inflammation.

## 4. Conclusions

In this study, MC-LR subchronic exposure seriously affected *L. catesbeianus* tadpoles’ gill function and microbiome. After MC-LR exposure, the permeability of the gill epithelial barrier was increased, while the ability to inhibit pathogenic bacteria was decreased. The increase of epithelial cells of inner gill filaments leads to hypoxia stress. *Elizabethkingia* may be an important source of LPS in the gill tissue after MC-LR exposure. LPS infiltration activated the TLR4/TNF-α signaling pathway, which may be an important pathway when it comes to causing gill inflammation after MC-LR exposure. MC-LR exposure disrupted the balance of redox homeostasis and activated the apoptosis. In addition, MC-LR subchronic exposure drove the gill microbiome to dysbiosis. In summary, these findings clue the first evidence that amphibian gills are very sensitive to MC-LR pollution, which provide a new basis for accurately assessing the risks of MC-LR pollution to amphibians.

## 5. Materials and Methods

### 5.1. Ethics Statement

The study was conducted according to the guidelines of the Declaration of Helsinki and approved by the Ethics Committee of Anhui Normal University and renewed on 7 June 2022 (protocol code AHNU-ET2022051).

### 5.2. Rearing of Animals and Sample Collection

Tadpoles (*Lithobates catesbeianus*) were exposed to environmental reared conditions based on the method of He et al. (2022) [14]. Briefly, tadpoles (G26 stage, 10 days post fertilization) were divided into three exposure groups: negative control, 0.5, and 2 μg/L MC-LR group (MC-LR purity > 95.0%; Institute of Hydrobiology, Chinese Academy of Sciences, Wuhan, China). Actual concentrations of MC-LR were 0, 0.5, and 1.95 μg/L, as determined by ELISA kits (a minimum detectable concentration limit of 0.1 ng/mL, Institute of Hydrobiology, Chinese Academy of Sciences, Wuhan, China) throughout the exposure period. As the measured concentrations were very close to the nominal concentrations, references to nominal concentrations are made throughout. In the following content, control, 0.5, and 2 μg/L MC-LR groups correspond to 0MCG, 0.5MCG, and 2MCG, respectively. Three replicate tanks of each group were conducted with each tank containing 20 tadpoles in 5 L medium. After 30 days (40 days post-fertilization) of MC-LR exposure, tadpoles from each tank were anesthetized with 0.02% MS-222 and gill samples were collected.

### 5.3. Histopathology of Gill

Gill tissues were fixed in 4% paraformaldehyde for histopathological observation. After paraformaldehyde-fixed gill tissues were dehydrated in a graded ethanol series, cleared in xylene, paraffin-embedded, and cut to 5 μm thickness, then stained with hematoxylin-eosin (HE) for histological observation [33]. Fifteen complete histological sections were randomly selected from each gill tissue (*n* = 6 gill tissue per group) and photographed using light microscope (Olympus BX61, Tokyo, Japan), and the number of gill filament epithelial cells and eosinophils were counted using Image-Pro Plus 6.0 software.

### 5.4. Biochemical Parameters of Gill

Each biological replicate (*n* = 3 replicates per group) consisted of three gills from each tank pooled together for biochemical analysis. Homogenized gill tissues were centrifuged (4° C, 15 min, 6000× *g*) to collect supernatants, which were transferred to fresh tubes for further analysis. Commercial colorimetric kits were used to detect the levels of glutathione (GSH) and reactive oxygen species (ROS), as well as the activities of caspase 3, superoxide dismutase (SOD), and lysozyme (LZM). Moreover, the enzyme-linked immunosorbent assay (ELISA) kits were conducted to check the contents of immunoglobulins (IgA, IgM, and IgG), interleukin-1β (IL-1β), and lipopolysaccharide (LPS). All kits were used following the manufacturer’s instructions (Nanjing Jiancheng Bioengineering Institute, Nanjing, China) with the biochemical measures normalized to the protein content. The optical density measurement was performed using a Tecan Spark 10M microplate reader (Tecan Trading AG, Zurich, Switzerland).

### 5.5. Antibacterial Activity of the Gill

Each biological replicate (*n* = 3 replicates per group) consisted of three gills from each tank pooled together for antimicrobial assays. Potential changes in gill antibacterial capacity induced by MC-LR exposure were assessed by disc diffusion assays [30,49]. Three pathogenic bacteria (e.g., *Escherichia coli*, CMCC44102; *Pseudomonas aeruginosa*, CICC 21625; and *Staphylococcus aureus*, CMCC26003) were used.

### 5.6. Transcriptome Sequencing and Bioinformatic Analyses

Each biological replicate (*n* = 3 replicates per group) consisted of three gills from each tank pooled together for transcriptome sequencing. TRIzol reagent (Invitrogen, Carlsbad, CA, USA) was used to extract total RNA, while total RNA quality was analyzed with an Agilent 2100 Bioanalyzer (Agilent Technologies, Santa Clara, CA, USA) and 9 cDNA libraries were prepared using the TruSeq^®^ RNA Sample Preparation Kit (Illumina, San Diego, CA, USA). And then performed sequencing on an Illumina Novaseq™ 6000 at the LC Sciences (Hangzhou, China). We first used the Cutadapt software (version 1.18) to remove the joint sequences and low-quality sequences in the original data to obtain clean data. Clean data was compared to the genomic data (*L. catesbeianus*, assembly RCv2.1) using HISAT2 software (version 2.1.0), and then data were assembled with StringTie software (v2.1.1). Assembly annotations were obtained using gffcompare software (version: gffcompare-0.9.8. Linux-x86-64) by comparing transcripts with the reference genome. The expression levels of all transcripts and genes were determined by calculating the fragments per kilobase per million reads mapped (FPKM) using StringTie and ballgown. Differentially expressed genes (DEGs) were determined with the judgment criteria of a fold change >2 or <0.5 and adjusted *p* ≤ 0.05 by the edgeR in R package. Then, Gene Ontology (GO) and Kyoto Encyclopedia of Genes and Genomes (KEGG) enrichments of the DEGs were analyzed. The bioinformatics analyses were performed as previously published protocols [14].

### 5.7. 16S rRNA Amplicon Sequencing and Bioinformatics

Each biological replicate (*n* = 3 replicates per group) consisted of three gills from each tank pooled together for 16S rRNA gene sequencing analysis. Total genomic DNA was extracted from the soil using the Mobio PowerSoil DNA Isolation Kit according to the manufacturer’s instructions (Mobio, Laboratories, Carlsbad, CA, USA). After genomic DNA extraction, the extracted genomic DNA was detected by 1% agarose gel electrophoresis. The 16S rDNA V3-V4 region was amplified with the primer sequences (338F, ACTCCTACGGGAGGCAGCAG and 806R, GGACTACHVGGGTWTCTAAT). Sequencing of the amplified fragment was performed on -the Illumina MiSeq platform (Illumina, San Diego, CA, USA). The paired-end reads obtained from MiSeq sequencing were spliced based on overlap relationships, while the sequence quality was quality controlled and filtered, then analyzed using QIIME2, and the reads were clustered into operational taxonomic units (OTUs) using Uparse with 97% similarity. The taxonomy of each OTU representative sequence was analyzed by RDP Classifier version 2.1.1 against the 16S rRNA database (Silva v138) using a confidence threshold of 0.7. The alpha diversity indices including Sobs, Chao, Shannon, and Simpson at OTU level were calculated by MOTHUR (v1.31.2). The beta-diversity based on the OTU level was calculated by measuring the Bray-Curtis distance using QIIME (version 1.9.1) and visualized using principal coordinate analysis (PCoA). Significance among samples were determined with analysis of similarity (ANOSIM) algorithm (9999 permutations) implemented using the Vegan package in R.

### 5.8. Statistical Analyses

All experimental data were analyzed statistically using R software version 3.6.3 (R Project for Statistical Computing, Vienna, Austria). Statistical comparisons among multiple groups were carried out by one-way ANOVA followed by the least significant difference (LSD) test (data with normal distribution and homogeneity of variance) or the Kruskal-Wallis test followed by the Dunn’s test (data not normally distributed or without homogeneity of variance). The correlation between the abundance of gill microbiota genera and inflammatory events indicators by LPS (*TLR4, IRAK4, IKKA, NF-κB, TNF-α, CLDN1* gene expression levels, and LPS, IL-1β content) was determined by Pearson analysis. *p*-value indicates a significant difference and *p* < 0.05 indicates statistical significance.

## Figures and Tables

**Figure 1 toxins-14-00479-f001:**
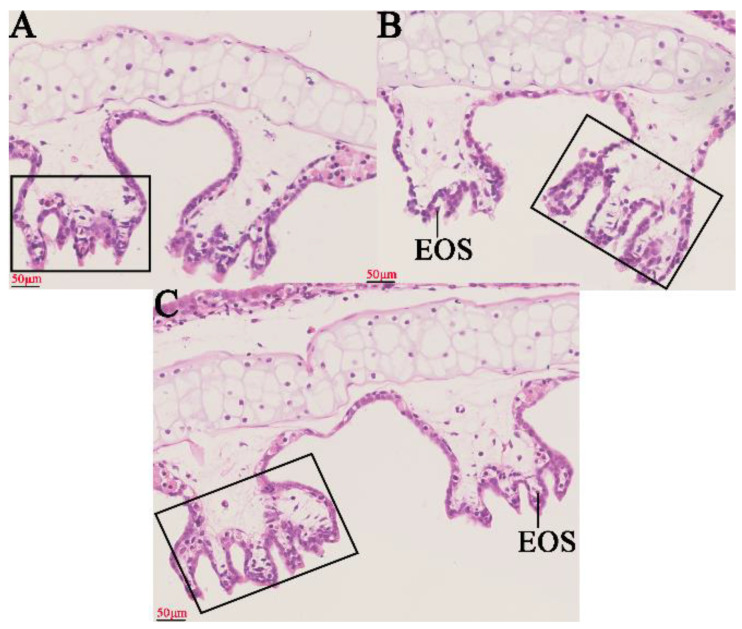
Pathological changes of gill tissue after exposure to 0 μg/L MC-LR (Control), 0.5, and 2 μg/L MC-LR. 0 μg/L MC-LR (Control) group (**A**); 0.5 μg/L MC-LR group (**B**); 2 μg/L MC-LR group (**C**). Magnifications: ×400 (**A**–**C**). Abbreviations: EOS, Eosinophils. Gill filament epithelial cells are shown within the solid rectangle.

**Figure 2 toxins-14-00479-f002:**
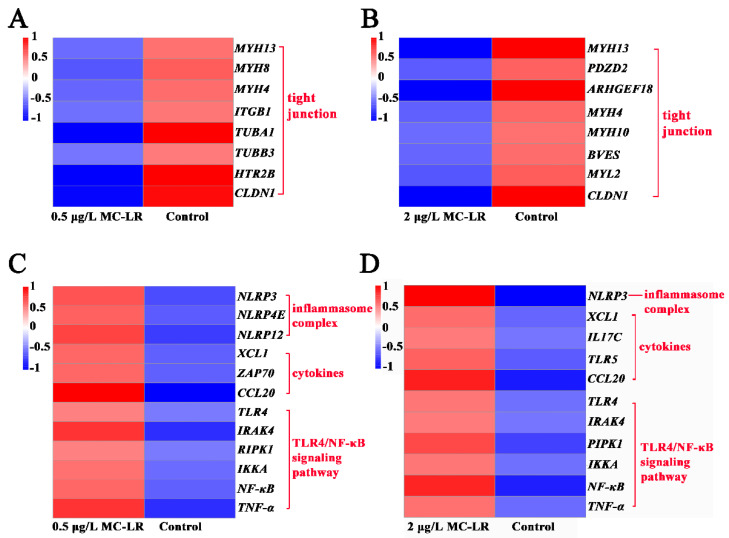
Heatmap of DEGs associated with tight junctions and inflammation in gill tissue after exposure to 0 μg/L MC-LR (Control), 0.5, 2 μg/L MC-LR. Control vs. 0.5 μg/L MC-LR group epithelium barriers (**A**); Control vs. 2 μg/L MC-LR group epithelium barriers (**B**); Control vs. 0.5 μg/L MC-LR group inflammation (**C**); Control vs. 2 μg/L MC-LR group inflammation (**D**).

**Figure 3 toxins-14-00479-f003:**
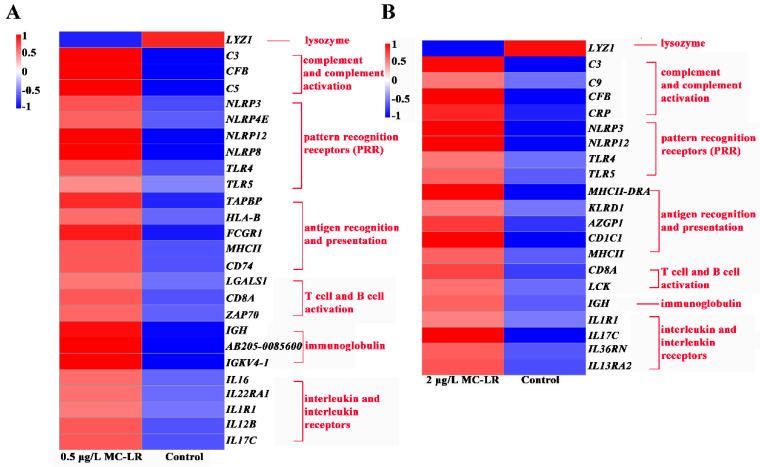
Heatmap of DEGs associated with immunity in gill tissue after exposure to 0 μg/L MC-LR (Control), 0.5, and 2 μg/L MC-LR. Control vs. 0.5 μg/L MC-LR group (**A**); Control vs. 2 μg/L MC-LR group (**B**).

**Figure 4 toxins-14-00479-f004:**
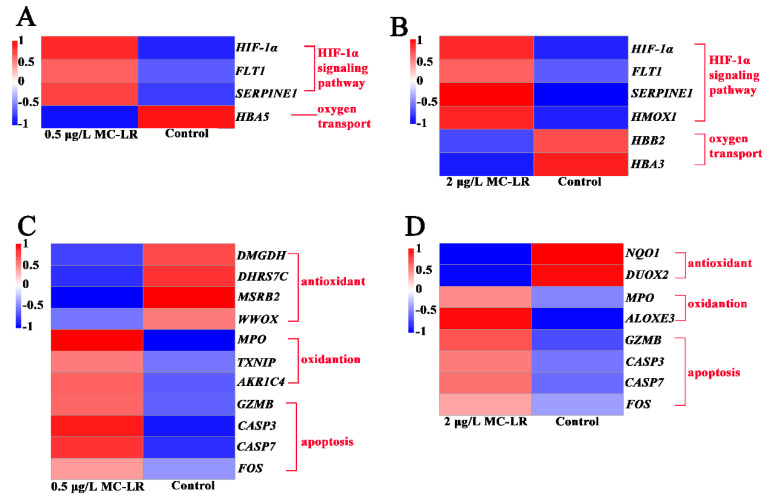
Heatmap of DEGs associated with hypoxic stress, oxygen transport, oxidative stress, and apoptosis in gill tissues after exposure to Control, 0.5, and 2 μg/L MC-LR. Hypoxic stress and oxygen transport at Control vs. 0.5 μg/L MC-LR group (**A**); Hypoxic stress and oxygen transport at Control vs. 2 μg/L MC-LR group (**B**); Oxidative stress and apoptosis at Control vs. 0.5 μg/L MC-LR group (**C**); Oxidative stress and apoptosis at Control vs. 2 μg/L MC-LR group (**D**).

**Figure 5 toxins-14-00479-f005:**
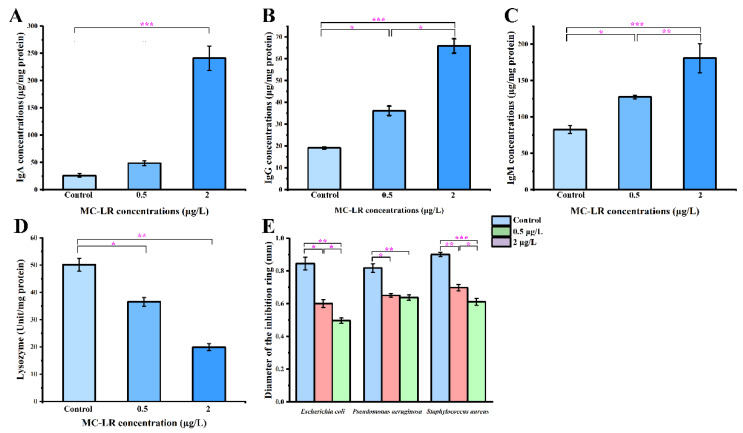
Alterations in IgA concentrations (**A**), IgG Concentrations (**B**), IgM concentrations (**C**), enzyme activities of lysozyme (**D**), the diameter of inhibition bacterial ring (**E**) among gill tissue after exposure to Control, 0.5, and 2 μg/L MC-LR. Values are presented as mean ± SEM of three replicates (*n* = 3). * *p* < 0.05, ** *p* < 0.01, and *** *p* < 0.001 indicate significant difference.

**Figure 6 toxins-14-00479-f006:**
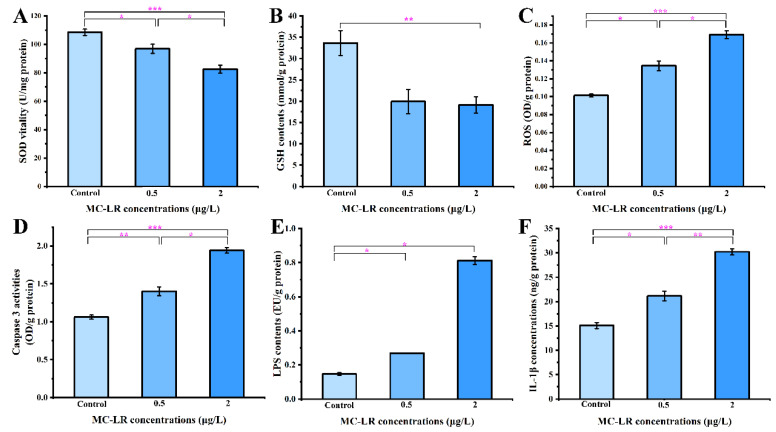
Alterations in SOD activities (**A**), GSH contents (**B**), ROS levels (**C**), Caspase 3 activities (**D**), LPS concentrations (**E**), and IL-1β concentrations (**F**) in gill tissues after exposure to Control, 0.5, and 2 μg/L MC-LR. Values are presented as mean ± SEM of three replicates (*n* = 3). * *p* < 0.05, ** *p* < 0.01, and *** *p* < 0.001 indicate significant difference.

**Figure 7 toxins-14-00479-f007:**
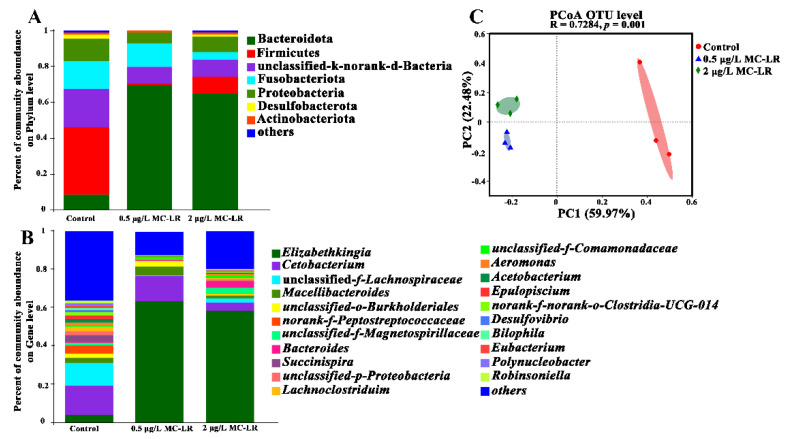
Distribution of microbial community at phylum level among gill tissues after exposure to 0 μg/L MC-LR (Control), 0.5, and 2 μg/L MC-LR (**A**). Distribution of microbial community at genus level among gill tissue after exposure to control, 0.5, and 2 μg/L MC-LR (**B**). PCoA ordination based on Bray–Curtis similarities of bacterial communities (**C**).

**Figure 8 toxins-14-00479-f008:**
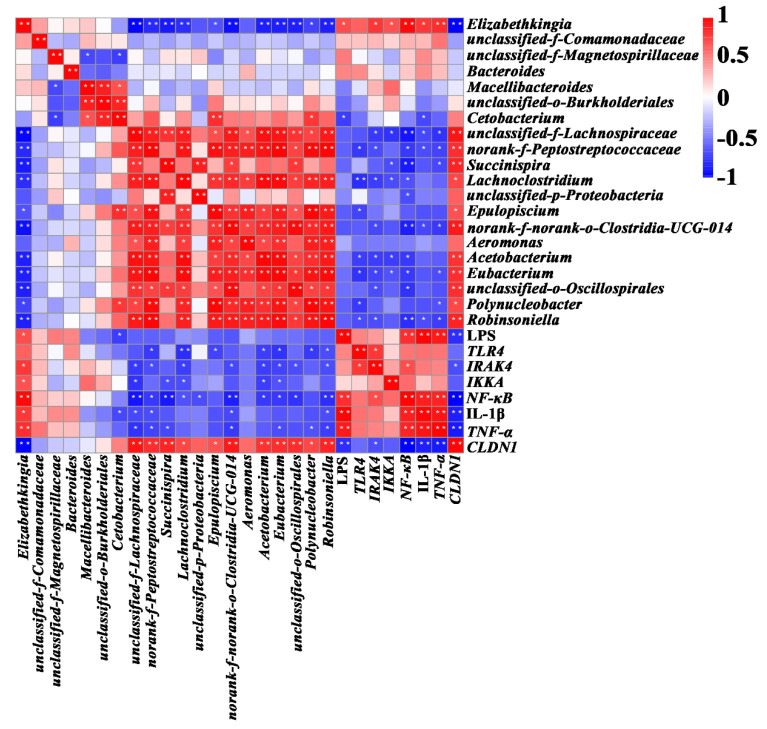
Correlation between gill microbial genera and metabolites. Blue and red colors indicate negative and positive correlations, respectively. Color intensity is proportional to extent of change. Significant correlations are indicated by asterisk (* *p* < 0.05 and ** *p* < 0.01; *r* > 0.5 or < −0.5).

## Data Availability

In this study, sequencing of 16S rRNA genes from frog tadpole gill microbiota samples has been submitted to the NCBI Sequence Read Archive under accession numbers PRJNA843355. The raw transcriptome data have been submitted to the NCBI. Sequence Read Archive under accession PRJNA843352.

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
