# Peer review of "Gill Junction Injury and Microbial Disorders Induced by Microcystin-Leucine Arginine in Lithobates catesbeianus Tadpoles"

_toxins, 2022, doi:10.3390/toxins14070479_

Round 1

Reviewer 1 Report

The present study describes the evaluation of the biological effects on the physiology of gills of tadpoles (Lithobates catesbeianus) submitted to environmentally realistic concentrations of microcystin MC-LR, a potent hepatotoxin produced by the cyanobacterium Microcystis aeruginosa. The tadpoles were exposed to concentrations of 0, 0.5, and 2 μg/L MC-LR for 30 days. After the treatment, the samples were submitted to histopathology analysis; biochemical analyses (SOD, LZM, and Caspase 3 activities, GSH contents, ROS levels, LPS, and IL-1β concentrations); antibacterial activity evaluation; transcriptome and 16S rRNA amplicon sequencing and bioinformatics. The methodology is adequate and was carried out in a competent manner. The results are interesting and indicate deleterious effects on the gills of tadpoles exposed to MC-LR, including gill inflammation, increased permeability, and reduced ability to inhibit pathogenic bacteria. The work represents an important contribution to this field of study and its publication is recommended. Before publication, it is recommended to make some adjustments to the manuscript. 

Please change:

Figure 1. ... 0.5 μg/L group (A) to … 0 μg/L group (A).

Line 276. Lysozyme is the widely distributed enzyme of innate immune system that antibacterial activity to Lysozyme is the widely distributed enzyme of innate immune system that exhibits antibacterial activity.

Figure 7A (y-axis title). Percentage of community abundance on genus level to phylum level.

Author Response

Dear Reviewer 1,

Thank you for your decision letter concerning our manuscript entitled “Gill Junction Injury and Microbial Disorders Induced by Microcystin-Leucine Arginine in Lithobates catesbeianus Tadpoles”, and your time regarding for our revision. I also appreciate all the critical comments from you and reviewers. We have carefully considered the comments and revised the manuscript accordingly. We marked the changes in red font in the marked revised manuscript. With these improvements, we hope that the current version can meet the Journal’s standards for publication. The following is a point-by-point response to all those comments and a list of changes we have made to the manuscript.

Point-by-point responses to the comments of the Editor and reviewers:

(The comments of the Editor and reviewers are in italics, which are followed by our responses.)

Q1. Figure 1. ... 0.5 μg/L group (A) to … 0 μg/L group (A).

A1: Thanks for your reminder. We have corrected the content in the revised manuscript, as following, “Pathological changes of gill tissue after exposure to 0, 0.5, and 2 μg/L MC-LR. 0 μg/L group (A); 0.5 μg/L MC-LR group (B); 2 μg/L MC-LR group (C). Magnifications: ×400 (A–C).”.

Q2 Line 276. Lysozyme is the widely distributed enzyme of innate immune system that antibacterial activity to Lysozyme is the widely distributed enzyme of innate immune system that exhibits antibacterial activity.

A2: Thanks for your kind reminder. The sentence has changed following your advice in the revised manuscript.

Q3 Figure 7A (y-axis title). Percentage of community abundance on genus level to phylum level.

A3: Thanks for your reminder. We have corrected Fig. 7A in the revised manuscript.

Reviewer 2 Report

The manuscript describes the effect of MC-LR on gill of Lithobates catesbeianus tadpoles. The topic is quite interesting, but I found a few things which should be improved.

1. MC-LR concentration should be monitored during 30-d exposure by LC-MS-MS method to confirm its exposure level in each treatment (0-2 ug/L).

2. The method part including data analysis is not described adequately to ensure the experiments can be repeated. Link to previous papers does not work well in that.

3. I am not sure if I understand well the heatmaps with transcriptomic data (Fig. 2-4). Log2 fold change is the ratio of the gene expression of MC-LR treated variants to non-treated ones (control), right? How can be the untreated variant affected as expressed in the heatmaps in the opposite way as the treated samples? The authors should explain this. I would prefer showing treated variants in one heatmap for all gene groups.

4. Figure 8 - It is not clear how were these correlations calculated, no description in the methods. How were the changes calculated for e.g., bacterial communities? The authors should explain that more to prevent misinterpretations.

5. The labels in the graphs are not easily readable.

Author Response

Dear Reviewer 2,

Thank you for your decision letter concerning our manuscript entitled “Gill Junction Injury and Microbial Disorders Induced by Microcystin-Leucine Arginine in Lithobates catesbeianus Tadpoles”, and your time regarding for our revision. I also appreciate all the critical comments from you and reviewers. We have carefully considered the comments and revised the manuscript accordingly. We marked the changes in red font in the marked revised manuscript. With these improvements, we hope that the current version can meet the Journal’s standards for publication. The following is a point-by-point response to all those comments and a list of changes we have made to the manuscript.

Point-by-point responses to the comments of the Editor and reviewers:

(The comments of the Editor and reviewers are in italics, which are followed by our responses.)

Q1: MC-LR concentration should be monitored during 30-d exposure by LC-MS-MS method to confirm its exposure level in each treatment (0-2 ug/L).

A1: Thanks for your advice. Actual concentrations of MC-LR were 0, 0.5 and 1.95 μg/L as determined by LC-MS-MS method calibrated enzyme-linked immunosorbent assay (ELISA) kit (a minimum detectable concentration limit of 0.1 ng/mL, Institute of Hydrobiology, Chinese Academy of Sciences, Wuhan, China) throughout the exposure period. As the measured concentrations were very close to the nominal concentrations, references to nominal concentrations are made throughout. The description has changed following your advice in the revised manuscript. A description has been added to the revised manuscript material methods.

Q2: The method part including data analysis is not described adequately to ensure the experiments can be repeated. Link to previous papers does not work well in that.

A2: Thanks for your suggestion. In the part of Transcriptome sequencing and 16S rRNA amplicon sequencing and bioinformatics, we have added a specific description in the revised manuscript.

We firstly used the Cutadapt software to remove the joint sequences and low-quality sequences in the original data to obtain the clean data. Clean data was compared to the genomic data (L. catesbeianus, assembly RCv2.1) using HISAT2 software, and then data were assembled with StringTie software. Assembly annotations were obtained using gffcompare software by comparing transcripts with the reference genome. The expression levels of all transcripts and genes were determined by calculating the fragments per kilobase per million reads mapped (FPKM) using StringTie and ballgown. Differentially expressed genes (DEGs) were determined with the judgment criteria of a fold change >2 or <0.5 and adjusted p ≤ 0.05 by the edgeR in R package. Then, Gene Ontology (GO) and Kyoto Encyclopedia of Genes and Genomes (KEGG) enrichments of the DEGs were analyzed.

The paired-end reads obtained from MiSeq sequencing were spliced based on overlap relationships, while the sequence quality was quality controlled and filtered, then analyzed using QIIME2, and the reads were clustered into operational taxonomic units (OTUs) using Uparse with 97% similarity. The taxonomy of each OTU representative sequence was analyzed by RDP Classifier version 2.1.1 against the 16S rRNA database (Silva v138) using a confidence threshold of 0.7. The alpha diversity indices including Sobs, Chao, Shannon, and Simpson at OTU level were calculated by MOTHUR (v1.31.2). The beta-diversity based on the OTU level was calculated by measuring the Bray- Curtis distance using QIIME (version 1.9.1), and visualized using principal coordinate analysis (PCoA). Significance among samples were determined with analysis of similarity (ANOSIM) algorithm (9999 permutations) implemented using the Vegan package in R.

Q3: I am not sure if I understand well the heatmaps with transcriptomic data (Fig. 2-4). Log2 fold change is the ratio of the gene expression of MC-LR treated variants to non-treated ones (control), right? How can be the untreated variant affected as expressed in the heatmaps in the opposite way as the treated samples? The authors should explain this. I would prefer showing treated variants in one heatmap for all gene groups.

A3: Thanks for your suggestions. the expression levels of all transcripts and genes were determined by calculating the fragments per kilobase per million reads mapped (FPKM) using StringTie and Ballgown (http://www.bioconductor.org/packages/release/bioc/html/ballgown.html). Differentially expressed genes (DEGs) with a fold change >2 or <0.5 and adjusted p ≤ 0.05 were selected. The fold change >2 indicates up regulation of gene expression relative to the control group. The fold change <0.5 indicates down regulation of gene expression relative to the control group. This is a common screening criterion for DEGs. The previous description may not be too clear and has been re-described in the revised manuscript. We used the average FPKM of DEGs in the treated and control groups for heatmaps. Two group DEGs were drawn together for a more intuitive comparison. However, since the DEGs of the two treated groups were not completely consistent, we chose to draw them separately.

Q4: Figure 8 - It is not clear how were these correlations calculated, no description in the methods. How were the changes calculated for e.g., bacterial communities? The authors should explain that more to prevent misinterpretations.

A4: Thanks for your suggestions, the correlation between the abundance of gill microbiota genera and inflammatory events indicators by LPS (TLR4, IRAK4, IKKA, NF-κB, TNF-α, CLDN1 gene expression levels and LPS, IL-1β content) was determined by Pearson analysis. The previous description may not have been very clear and has been redescribed in the revised manuscript. The calculation method of microbiota genera abundance has been added in the materials and methods section.

Q5: The labels in the graphs are not easily readable.

A5: Thanks for the reminder, we have adjusted it in the manuscript.

Reviewer 3 Report

The number of publications that study various issues related to the negative manifestations of the mass development of cyanobacteria in water bodies is steadily increasing. Toxic blooms of cyanobacteria are a global problem, and they significantly deteriorate the quality of drinking water, causing numerous negative and irreversible processes in water bodies, including the degradation of aquatic ecosystems. However, detailed studies on the effects of toxins on aquatic organisms are still insufficient. The presented study successfully fills this gap. The methods used and parameters obtained allowed the authors to evaluate the effect of МС-LR in two concentrations (0.5, and 2 μg/L) on tadpoles (Lithobates catesbeianus). The range of methods used is rather wide: cultivation, histopathological observation, biochemical analysis, antimicrobial assays, transcriptome and 16S rRNA amplicon sequencing, and bioinformatic analyses. The results obtained reliably revealed substantial impacts of MC-LR exposure on gill of amphibian. The authors presented their study conscientiously and convinced me that the tadpole gill function is greatly affected by exposure to MC-LR, even at low concentrations.   

I have a comment to the manuscript. The authors use the expression “realistic concentrations of 0, 0.5, and 2 μg/L MC-LR”, but the concentration of 0 μg/L MC-LR is just a negative control. I think that this group should be designated simply as a control. 

It is obvious that the authors have long and successfully dealt with this topic. A very similar article was published in 2022 and is cited by the authors (Jun He  et al. Microcystin-leucine arginine exposure induced intestinal lipid accumulation and MC-LR efflux disorder in Lithobates catesbeianus tadpoles. Toxicology. 2022 Jan. 15;465:153058.  doi: 10.1016/j.tox.2021.153058. Epub 2021 Dec 2.). The article is devoted to the study of amphibian intestines. The submitted article should be supplemented with publications of other authors on this topic, including those publications that do not confirm the effect of MC-LR on amphibians. For example, Zhang, Xiaoming & Zhang, Pengcheng & He, Jiawan & Liu, Yongding & Matsuura, Hiroshi & Watanabe, Makoto. (2012). Grazing on toxic cyanobacterial blooms by tadpoles of edible frog Rana grylio. Phycological Research. 60. 10.1111/j.1440-1835.2011.00627.x, Zikov´a, A., Lorenz, C., Lutz, I., Pflugmacher, S., Kloas, W., 2013. Physiological responses
of Xenopus laevis tadpoles exposed to cyanobacterial biomass containing
microcystin-LR. Aquat. Toxicol. 128-129, 25–33.

There are some remarks to the text design:

1)     Make the photos with high magnifications in Figure 1. Even with the zoom, it is difficult to see the details of the figure. The authors mark eosinophils, but they are not visible. It would be better to make an insertion with high magnification in this figure. 

2)     The font of the legend in Figure 7 is very small, and nothing prevents it from being made larger.

Author Response

Dear Reviewer 3,

Thank you for your decision letter concerning our manuscript entitled “Gill Junction Injury and Microbial Disorders Induced by Microcystin-Leucine Arginine in Lithobates catesbeianus Tadpoles”, and your time regarding for our revision. I also appreciate all the critical comments from you and reviewers. We have carefully considered the comments and revised the manuscript accordingly. We marked the changes in red font in the marked revised manuscript. With these improvements, we hope that the current version can meet the Journal’s standards for publication. The following is a point-by-point response to all those comments and a list of changes we have made to the manuscript.

Point-by-point responses to the comments of the Editor and reviewers:

(The comments of the Editor and reviewers are in italics, which are followed by our responses.)

Q1: I have a comment to the manuscript. The authors use the expression “realistic concentrations of 0, 0.5, and 2 μg/L MC-LR”, but the concentration of 0 μg/L MC-LR is just a negative control. I think that this group should be designated simply as a control.

A1: Thanks for your suggestion, we have adjusted it in the revised manuscript, realistic concentrations of 0.5, and 2 μg/L MC-LR for the exposure group, and 0 μg/L MC-LR for a negative control.

Q2: It is obvious that the authors have long and successfully dealt with this topic. A very similar article was published in 2022 and is cited by the authors (Jun He et al. Microcystin-leucine arginine exposure induced intestinal lipid accumulation and MC-LR efflux disorder in Lithobates catesbeianus tadpoles. Toxicology. 2022 Jan. 15;465:153058.  doi: 10.1016/j.tox.2021.153058. Epub 2021 Dec 2.). The article is devoted to the study of amphibian intestines. The submitted article should be supplemented with publications of other authors on this topic, including those publications that do not confirm the effect of MC-LR on amphibians. For example, Zhang, Xiaoming & Zhang, Pengcheng & He, Jiawan & Liu, Yongding & Matsuura, Hiroshi & Watanabe, Makoto. (2012). Grazing on toxic cyanobacterial blooms by tadpoles of edible frog Rana grylio. Phycological Research. 60. 10.1111/j.1440-1835.2011.00627.x, Zikov´a, A., Lorenz, C., Lutz, I., Pflugmacher, S., Kloas, W., 2013. Physiological responses of Xenopus laevis tadpoles exposed to cyanobacterial biomass containing microcystin-LR. Aquat. Toxicol. 128-129, 25–33.

A2: Thank you for your advice. We have added these references in the discussion section.

Q3: Make the photos with high magnifications in Figure 1. Even with the zoom, it is difficult to see the details of the figure. The authors mark eosinophils, but they are not visible. It would be better to make an insertion with high magnification in this figure.

A3: Thanks for the reminder, we have adjusted it in the revised manuscript.

Q4: The font of the legend in Figure 7 is very small, and nothing prevents it from being made larger.

A4: Thanks for your kind reminder. We have changed Fig. 7 to make it easier to read.

Round 2

Reviewer 2 Report

The authors addressed my comments well and improved their manuscript.